# CreDes: Causal Reasoning Enhancement and Dual-End Searching for Solving Long-Range Reasoning Problems using LLMs

## Abstract

Large language models (LLMs) have demonstrated limitations in handling combinatorial optimization problems involving long-range reasoning, partially due to causal hallucinations and huge search space. As for causal hallucinations, i.e., the inconsistency between reasoning and corresponding state transition, this paper introduces the Causal Relationship Enhancement (CRE) mechanism combining cause-effect interventions and the Individual Treatment Effect (ITE) to guarantee the solid causal rightness between each step of reasoning and state transition. As for the long causal range and huge search space limiting the performances of existing models featuring single-direction search, a Dual-End Searching (DES) approach is proposed to seek solutions by simultaneously starting from both the initial and goal states on the causal probability tree. By integrating CRE and DES (CreDes), our model has realized simultaneous multi-step reasoning, circumventing the inefficiencies from cascading multiple one-step reasoning like the Chain-of-Thought (CoT). Experiments demonstrate that CreDes significantly outperforms existing State-Of-The-Art (SOTA) solutions in long-range reasoning tasks in terms of both accuracy and time efficiency.

## 1 Introduction

Reasoning aims to realize the causal transfer from the initial state to the goal state through several intermediate steps, which widely exists in the domains of Societal Simulation (Gandhi et al., 2024; Xu et al., 2024; Hua et al., 2023), Economic Simulation (Li et al., 2023a; Zhao et al., 2023; Xia et al., 2024), Game Theory (Xu et al., 2023b; Mao et al., 2023; Zhang et al., 2024) and Gaming (Mukobi et al., 2023; Huang et al., 2024; Shao et al., 2024), etc. LLMs like GPT-3 have shown competitive performances in many reasoning tasks (Brown et al., 2020; Chowdhery et al., 2023; Betker et al., 2023). However, their performances and efficiency are limited when dealing with complex combinatorial optimization problems that require multi-step long-range reasoning (Kaddour et al., 2023).

The first challenge is causal hallucinations, i.e., causality between one-step reasoning (OSR) and state transition in LLMs is not always guaranteed. Similar to pre-trained LLMs that are prone to produce hallucinations when processing certain factual information, causal hallucinations reflect the fact that LLMs lack rigor due to inherent randomness in accomplishing complex mathematical (Cobbe et al., 2021b; Imani et al., 2023; Lewkowycz et al., 2022), logical (Liu et al., 2023; Xu et al., 2023a), or common-sense reasoning (Zhao et al., 2024a; Sharan et al., 2023; Xenos et al., 2024), which is somehow entrenched in statistical inevitability and independent of the Transformer architecture or data quality (Kalai & Vempala, 2023). For example, CoT-based finite-step reasoning methods (Wei et al., 2022b; Zheng et al., 2023) suffer from causal hallucinations, which cannot effectively ensure the causality between OSR and state transition in LLMs, resulting in unreliable reasoning and relatively low success rates (especially for long-range reasoning problems with significant error accumulation effects). The reasonableness between OSR and state transition can be summarized as follows: There is a causal relationship between reasonable OSR and state transition. However, for unreasonable OSR, there is only a correlation or no relationship with state transition. This suggests that training solely with cross-entropy loss, as commonly used in most methods, does not address the model's causal rigor well enough. Inspired by this, we designed the CRE mecha-

nism to make each step of reasoning correct and *causally sound* by embedding the causality measure between OSR and state transition into the training loss, thus more closely modeling the rigor, adaptability, and comprehensiveness of human reasoning (Bao et al., 2024).

The second challenge is that long-range reasoning problems have a huge search space. Although complex architectures such as CoT, Tree of Thought (ToT) (Yao et al., 2024), and Program of Thought (PoT) (Chen et al., 2022) can effectively improve the reasoning accuracy of LLMs through external guidance, they are limited when handling long-range reasoning processes and task decomposition. A crucial reason is that long-range reasoning has a huge state space, i.e., each branch in the state transition process expands the search space approximately exponentially. Most of the existing LLM-based methods, e.g., Monte Carlo search (Zhao et al., 2024b), are based on unidirectional reasoning, making them time-inefficient and easy to fall into local optima when dealing with reasoning problems with large search spaces. In this paper, a bi-directional Dual-End Searching method is developed, which first decomposes a long-range reasoning problem into a combination of short-range reasoning problems and then searches for the intersection of two causal probability trees starting from the initial and goal states, respectively.

A structured and general reasoning framework, CreDes, is developed for long-range reasoning with LLMs in this paper, and the contributions can be summarized as follows:

**First, the CRE mechanism is introduced to improve the rigor of LLM-based long-range reasoning methods:** Structural Causal Modeling (SCM) is exploited to enhance the causality between OSR and state transitions, involving performing causal interventions and optimizing ITE during training, which has effectively alleviated causal hallucinations in long-range reasoning of LLMs.

**Second, the DES method is developed to improve the search efficiency for long-range reasoning:** After constructing causal probability trees starting from the initial states and ending at the goal states, long-range reasoning (e.g., 12 steps) is divided into more manageable combinations of smaller segments (e.g., 2 or 4 steps) by bi-directional approaching. The final reasoning paths are selected by constructing a new metric guaranteeing both low reasoning hallucination and high reasoning quality. By avoiding long-range sequential search from scratch, the DES method has dramatically lowered the complexity when solving long-range reasoning problems.

**Third, simultaneous multi-step reasoning is realized to improve the time-efficiency of long-range reasoning:** By integrating CRE and DES, CreDes can perform simultaneous multi-step reasoning within the model, i.e., avoiding the inefficiency of *cascading single-step reasoning* in frameworks such as CoT. While ensuring the accuracy of the reasoning process, CreDes can significantly reduce the time required for multi-step reasoning in LLMs.

**Fourth, adequate and rigorous testing of CreDes:** CreDes has been extensively tested in the Blocksworld, GSM8K, and Hanoi Tower scenarios, respectively, and the experimental results show that CreDes outperforms existing SOTA regarding reasoning accuracy and time efficiency.

## 2 RELATED WORK

**Decision-Making Capabilities in LLMs:** The core of intelligence partially lies in planning, which encompasses generating a sequence of actions aimed at accomplishing a predefined objective (McCarthy et al., 1963; Bylander, 1994). Classical planning methods have found extensive application in robotics and embodied environments, where they are commonly employed to guide decision-making processes externally (Camacho & Bordons, 1999; Jiang et al., 2019). Recent advancements, such as the Chain-of-Thought model (Wei et al., 2022b; Kojima et al., 2022; Chu et al., 2023), have significantly bolstered the LLMs' capability to perform detailed reasoning (Huang et al., 2022; Singh et al., 2023; Ding et al., 2023). This model breaks down intricate queries into a series of manageable steps, thereby enhancing the LLMs' decision-making ability. Subsequent initiatives like ReACT (Yao et al., 2022) have modified this approach to improve reasoning ability in decision contexts using a CoT-based framework. Additionally, Reflexion (Shinn et al., 2024) provides a corrective mechanism that enables LLMs to recognize their errors during the decision-making process, reflect on these mistakes, and make accurate decisions in subsequent attempts. Further developments have led to the creation of tree-based decision-making frameworks that tailor LLM capabilities to specific scenarios. The Tree-of-Thought (Yao et al., 2024) utilizes Breadth First Search (BFS) and Depth First Search (DFS) algorithms to facilitate decision-making in activities such as the Game of 24, Cre-

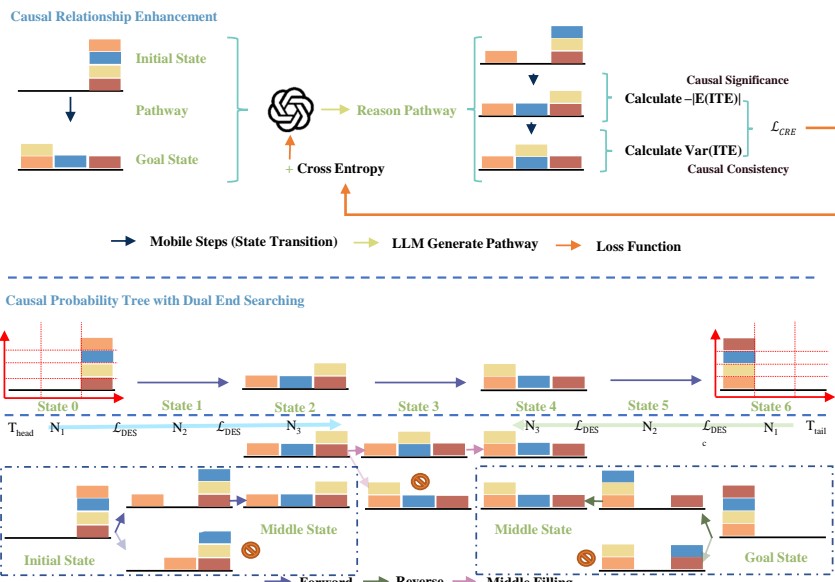

Figure 1: Integrating Causal Relationship Enhancement (CRE) and Dual-End Searching (DES).

ative Writing, and Mini Crosswords. Meanwhile, Reasoning via Planning (RAP) (Hao et al., 2023) employs the Monte Carlo Tree Search technique to optimize solutions across tasks like Blocksworld (Valmeekam et al., 2024), Math Reasoning (Zhu et al., 2022). DFSDT (Qin et al., 2023) proposed an efficient version of DFS for LLMs to make decisions, but it lacks the judgment ability to evaluate different decisions. JUDEC (Ye et al., 2023) utilizes an Elo rating system to enable LLMs to develop self-assessment capabilities, thereby enabling them to generate optimal solutions for a wide range of real-world tasks, independent of any task-specific expertise. Lastly, Graph-of-Thought (Yao et al., 2023) represents the thoughts as nodes in a graph, combining thoughts non-sequentially. All of the above work shows that LLM has excellent potential for handling long-range reasoning tasks and shows some advantages in areas such as inference tasks.

**Integrating Causal Analysis in LLMs for Multi-step Decision-Making:** Causal analysis aims to discern and elucidate the causal relationships between actions, circumstances, or decisions. This method entails investigating the origins or causes leading to an event and the potential consequences that follow (Heise, 1975; Imbens & Rubin, 2015; Feder et al., 2022). Although various causal models may produce identical observational distributions, they can yield distinct distributions when interventions are applied (Peters et al., 2017; Wang et al., 2024). Therefore, using interventions allows for the distinction of possible causal models that align with the observed data (Hagmayer et al., 2007; Pearl, 2009). This enhances the causal consistency and significance of the model training process. Previous work suggests that, while CoT has been lauded for its potential to improve task performance, its application does not always lead to enhanced outcomes (Kojima et al., 2022; Nichols et al., 2020). Also, research has shown that the statistical pretraining of LLMs encourages models to achieve high empirical performance but not necessarily to reason (Zhang et al., 2022; Turpin et al., 2024; Zečević et al., 2023; Lanham et al., 2023). Motivated by this, we designed the CRE mechanism combining causal analysis and LLMs to control the causal hallucinations when solving long-range reasoning problems.

**Solving Multi-step Problems with LLMs:** Recent studies have shown that with substantial design, LLMs are capable of performing not only basic arithmetic tasks but also complex multi-step reasoning (Power et al., 2022; Wei et al., 2022a). For instance, increasing computational resources significantly enhances the accuracy of datasets like GSM8K (Cobbe et al., 2021a). Concurrently, Research (Yang et al., 2023b) demonstrated that a 2B parameter LLM could achieve 89.9% accuracy in 5x5 multiplication tasks using curriculum learning with 50 million training instances. This evidence suggests that adequately scaled LLMs can process multiple reasoning steps effectively internally. While trees are frequently used to represent games (especially extensive-form games (Leonard, 2006; Leyton-Brown & Shoham, 2008)) and sequential reasoning problems (Rus-

sell & Norvig, 1995), it was Shafer's groundbreaking work (Shafer, 1996) that initially established a framework for understanding causality through the use of probability trees. However, it can also be inferred from Shafer's work that LLMs struggle with long-range reasoning problems involving multiple steps but excel in short-range reasoning tasks. This insight led to the development of DES, which breaks down the Long Range Reasoning Question into smaller parts and then searches for connection points from both the head and tail nodes by integrating causal probability trees.

# 3 METHOD

The pipeline of CreDes is illustrated in Fig. 1. It comprises two main components: CRE and DES. In CRE, the inputs of LLMs for training are the initial state, goal state, and pathway (containing a series of OSRs), while for testing, the inputs are the initial and goal states only. The DES starts from the initial and goal states of the probability tree, expands them into two intermediate states, and uses the CRE-trained model to infer the pathway between them, ultimately producing the complete pathway.

## 3.1 PROBLEM DEFINITION

To further improve the capability of LLMs in solving combinatorial optimization problems that involve a finite number of discrete intermediate steps, we conducted experiments using the Blocksworld and Hanoi Tower datasets with 7B parameter models. The Blocksworld dataset includes 602 test cases categorized by the minimum number of required actions, ranging from 2 to 12 steps. For Hanoi Tower, cases are grouped based on the complexity related to the number of disks and poles, which directly influences the solution steps.

For each category, our model is trained on 80 samples without common instructions. In the reasoning process, the following elements are included: initial state, OSR, state transition, next state, and goal state, as shown in Fig. 2. During testing, the model was tested on new, categorically similar samples from different datasets, assessing its ability to transform the initial state to the goal state successfully.

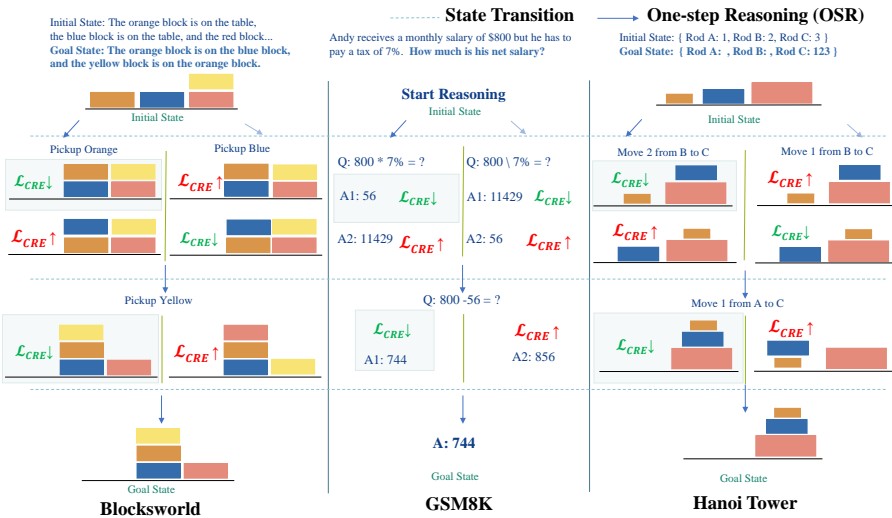

Figure 2: Schematic illustration of Causal Relationship Enhancement(CRE).

## 3.2 CAUSAL SIGNIFICANCE AND CONSISTENCY

In causal inference, ITE measures the difference in outcomes for an individual with and without a specific treatment. A larger ITE typically indicates a stronger causal relationship between random variables. Its definition is as follows:

$$\text{ITE}_i = Y_i(W = 1) - Y_i(W = 0) \tag{1}$$

where $Y_i(W = 1)$ and $Y_i(W = 0)$ are the potential treated/control outcomes of sample $i$. $W$ represents the treatment assignment. ITE is generally encouraged to be as large as possible, and prior work (Pearl, 2018) has used ITE as a discriminator of causality strength. The larger the ITE, the more significant the causality. However, we found that only enhancing the significance of causality through improving E(ITE) is not enough; improving the stability of causality by constraining Var(ITE) is indeed more critical.

As is shown in Fig.3, we conducted a statistical analysis of the distribution of the model's output results, which demonstrates that these outputs include various possibilities, such as true positives, false positives, and false negatives, as shown in the experimental results. Previous work has shown that large language models possess basic logical reasoning abilities, so we aim to enhance this capability rather than rebuild it. The model's responses follow a approximate normal distribution $\text{ITE}_i \sim N(\mu, \sigma^2)$ for repeated experiments on a single sample (Hartung & Knapp, 2001; Van der Elst et al., 2021; Lei & Candès, 2021). In this context, the mean of the normal distribution aligns with the causal significance for individual-level $\text{ITE}_i$, while the variance reflects the causal consistancy of individual effects. Based on this, we propose the following logical extension, as is shown in Fig.3.

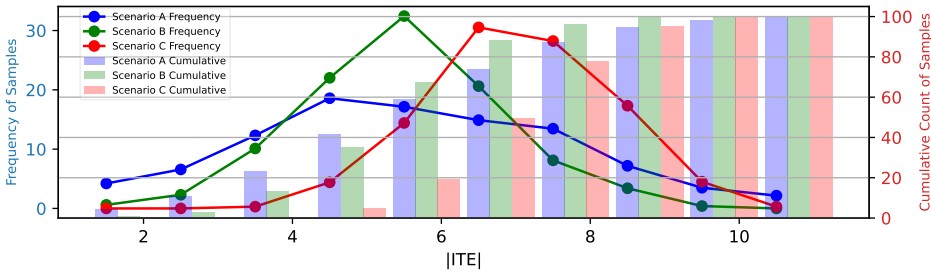

Figure 3: Exact Frequency and Cumulative Histogram for Scenarios A, B, and C. For the convenience of image drawing, the coordinate axes of this image have been scaled to a certain degree and do not represent the actual values.

In Scenario A, the model relies solely on Cross-Entropy, where causal relationships are determined without considering the significance or consistency of individual effects. Scenario B introduces Var(ITE) to enhance causal consistency, but variability among individual responses may still obscure the strength of the causal effect. Scenario C, the ideal outcome, is achieved by jointly optimizing both E(ITE) and Var(ITE), resulting in more significant and more reliable causal relationships.

In conclusion, we believe that both causal significance E(ITE) and causal consistency Var(ITE) contributes to the transition from Scenario A and B to Scenario C, a scenario that we expect to achieve, while leveraging Cross-Entropy to assure the model's correctness capabilities. By jointly controlling these three factors, we achieve improved model performance.

### 3.3 CAUSAL RELATIONSHIP ENHANCEMENT (CRE)

Firstly, all the samples are classified into two categories: Correct and Incorrect. Within the Incorrect category, three scenarios exist, i.e., a correct OSR leading to an incorrect state transition, an incorrect OSR leading to an incorrect state transition, and an incorrect OSR resulting in a correct state transition. Given this, it is evident that we need to strengthen the causal connection between the OSR and the transition, and reduce the occurrence of samples where the OSR and the transition are non-causal. In CRE, we first use the ITE to estimate the causality between OSR and state transition quantitatively, and then embed the $|E(\text{ITE})|$ and Var(ITE) into the loss function in the training process (the remaining is cross-entropy), enhancing the causality of state transitions. As is shown in Fig. 2 and the upper part of Fig. 1, we leave the reasoning path selection to be controlled by the cross-entropy loss, while the suppression of hallucinations is handled by the ITE loss. Perplexity (PPL) is a metric used to evaluate the performance of a LLM, indicating how well the model predicts

the next word in a sequence, and lower values signify better predictive accuracy. The estimation of ITE is detailed as the follows:

Given binary variables $X$ and $Y$ indicating the correctness of OSR and next state (state transition), respectively, i.e., $X, Y \sim B(0, 1)$, and $X = 1$ (or $Y = 1$) means correctness. First, we calculate the cause-effect interventions between $X$ and $Y$, then subsequently modify the distribution of $Y$ by intervening in $X$. From a statistical correlation perspective, if $X$ and $Y$ are correlated, $Y$ can be predicted using $X$. However, if there is no causal relationship between $X$ and $Y$, intervening in $X$ will not alter the distribution of $Y$. Hence, if $X$ and $Y$ are correlated but not causally linked, then manipulating or intervening in $X$ would not lead to any changes in the distribution of $Y$. This distinction is crucial in statistical analysis and experimental design because it addresses the potential fallacy that correlation inherently means causation.

Under the intervention, the proportion of positive and negative cases (hallucinations) in the model output samples remains roughly unchanged; the more significant the causal relationship between different OSRs and corresponding positive and negative cases, the lower the $-|\mathrm{E(ITE)}|$. The reason is that cross-entropy basically ensures the majority of positive cases. At the same time, lowing $\mathrm{Var(ITE)}$ reduces the occurrence of negative cases, making the distribution of positive and negative cases more stable, $\alpha$ and $\beta$ are dynamic coefficients fitted with the training process. Consequently, we incorporate the ITE into the loss function, as is shown in (2) and (3), $p_{1|X}$ and $p_{0|X}$ denote the conditional probabilities of $Y$ being *1* and *0*, respectively, given the state of $X$.

$$\mathcal{L}_{CrossEntropyLoss} = - \big[ Y \log(p_{1|X}) + (1 - Y) \log(p_{0|X}) \big] \tag{2}$$

$$\mathcal{L}_{CRE} = \mathcal{L}_{CrossEntropy} - \alpha|\mathrm{E(ITE)}| + \beta\mathrm{Var(ITE)} = \ln(\mathrm{PPL}) \tag{3}$$

### 3.4 CAUSAL PROBABILITY TREES WITH DUAL END SEARCHING (DES)

In this section, we improve the success rate of LLMs in solving long-range reasoning problems, such as the 12-step Blocksworld scenario, by leveraging their higher success rates in simpler 2-step and 4-step tasks. The main implementation process of DES is as follows:

**Step1:** We build two causal probability trees from the initial and goal states, with nodes representing reasoning states and arrows denoting causal relationships. These trees outline possible reasoning paths within a limited number of steps.

**Step2:** By matching their leaves, we identify end-to-end permutation schemes to form a continuous, and feasible path (as shown in Fig. 1 and Fig. 4). The DES framework ensures optimal path selection by expanding two trees from the head and tail ends ($T_{head}$ and $T_{tail}$). Probabilities are calculated based on the likelihood of reaching each state, with expansion directions chosen to reduce the distance to the target. This method balances exploration and exploitation, avoiding premature convergence to suboptimal solutions.

**Step3:** After several layers of expansion, the leaf nodes of the head and tail trees are matched, and a distance matrix $M$ is constructed. This matrix quantifies the spatial relationships between the end leaf nodes in $T_{head}$ and $T_{tail}$. The distance matrix is computed as the Euclidean distance between the coordinates of each leaf node in the two trees, as follows:

$$M_{ij} = \sqrt{(x_i - x_j)^2 + (y_i - y_j)^2} \tag{4}$$

where $(x_i, y_i)$ represents the coordinates of node $i$. As illustrated in Fig. 4, the distance matrix provides a quantitative representation of the separation between the nodes, helping guide the selection of the next expansion and pruning steps.

As the trees expand, the reasoning path is updated with the latest expansion results every four steps. This mechanism ensures that the bidirectional tree expansion from both the head and tail proceeds systematically, converging towards an optimal path. Moreover, this process retains flexibility, allowing changes in the expansion direction when needed to avoid being trapped in local minima.

**Step4:** To select the optimal expansion path, DES employs the $\mathcal{L}_{DES}$ metric, which balances between the length of the path and the correctness of the causal reasoning. Specifically, $\mathcal{L}_{DES}$ incorporates the Average Treatment Effect (ATE) to assess the causal relationship between tree expansion

and the resulting reduction in distance. The ATE is calculated as follows:

$$\text{ATE}(A) = E[\text{ITE}(A, B)] = E[E(A|do(B = 1)) - E(A|do(B = 0))] \quad (5)$$

where $A$ represents the reduction in distance $\delta D$ between two successive nodes $N_i$ and $N_{i-1}$, and $B$ is the number of layers in which the current leaf node is located. The distance reduction $\delta D$ is estimated by calculating the Euclidean distance between the current node and the target node. The metric used to optimize path selection is expressed as:

$$\mathcal{L}_{DES} = -|\text{ATE}(\delta D_{T_{head}}^{N_i - N_{i-1}})| - |\text{ATE}(\delta D_{T_{tail}}^{N_i - N_{i-1}})| + D \quad (6)$$

This function combines the ATE for both the head and tail trees, penalizing paths that deviate from the optimal causal direction while minimizing the total distance. The whole calculation process and execution details of DES can be found in Algorithm 1 and Fig. 4, which illustrate the complete workflow from tree generation, distance matrix construction, and path expansion to the final selection of the optimal path.

Algorithm 1: DES (Taking the 12-step Blocksworld as an example)

1: **Input:** $State_{init}$ and $State_{goal}$, denoting the initial and goal states
2: **Output:** Complete 12-step solution
3: Construct $T_{head}$ and $T_{tail}$ from $State_{init}$ and $State_{goal}$
4: Match leaves of $T_{head}$ and $T_{tail}$ to form paths
   - From $T_{head}$, infer 4 steps toward $T_{tail}$ based on reducing distance $D$.
   - Similarly, infer 4 steps from $T_{tail}$ toward $T_{head}$.
   - Calculate the Euclidean distances between the resulting end nodes of both trees to form a distance matrix $M$.
   - Select the three shortest distances and pass the corresponding node pairs to the model for further solving attempts.
5: **for** every four steps **do**
6:     Determine intermediate steps and fill in details
7: **end for**
8: **for** expanding $T_{head}$ and $T_{tail}$ **do**
9:     Calculate distance $D$
10:     Minimize $\mathcal{L}_{DES}$
11:     **if** local optimum detected **then**
12:         Assess alternative routes
13:     **end if**
14: **end for**

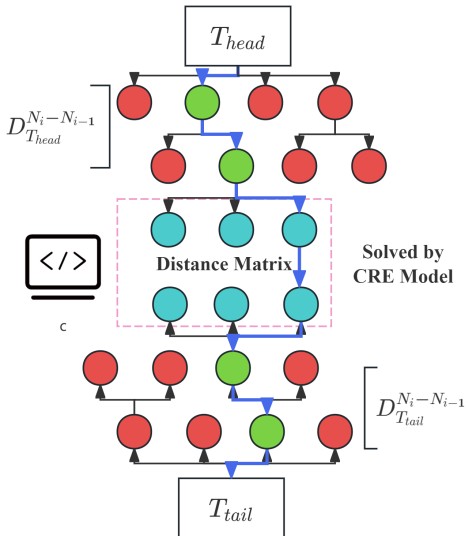

Figure 4: Sketch of the DES (in 9-steps)

Table 1: Accuracy under GSM8K

| Model | RAP | RoT | CoT | **CRE** |
|---|---|---|---|---|
| Llama-2-7B | 0.51 | 0.54 | 0.47 | **0.92** |
| Llama-2-13B | 0.50 | 0.57 | 0.49 | **0.93** |
| Phi-2-7B | 0.45 | 0.48 | 0.48 | **0.89** |
| Mistral-7B | 0.39 | 0.32 | 0.31 | **0.85** |
| Mixtral-8x7B | 0.48 | 0.50 | 0.49 | **0.90** |

During the expansion process of the probability trees at both ends, we intervene by minimally the change in the $\mathcal{L}_{\mathcal{DES}}$, directing the expansion toward our desired outcome. Minimizing $\mathcal{L}_{\mathcal{DES}}$ realizes the pruning and unfolding direction judgment, prioritizing the direction with the lowest $\mathcal{L}_{\mathcal{DES}}$ as the unfolding direction. The whole process of DES is in **Algorithm 1**.

## 4    EXPERIMENT

In this section, we validated the effectiveness of CreDes compared to baseline approaches.

## 4.1 SETUP

**Blocksworld:** There are $n$ blocks initially placed randomly on a table (Valmeekam et al., 2024). The LLM's goal is to stack these blocks in a specified order. The LLM can perform four actions: pick up a block from the table, put down a block it is holding onto the table, unstack a block from another to hold it, and stack the block in its hand onto another block. The LLM can only manipulate one block at a time, and blocks with others on top are immovable.

**GSM8K:** The GSM8K dataset (Cobbe et al., 2021a) includes 1,319 diverse grade school math word problems curated by human problem writers. These tasks typically begin with a description and culminate in a final question requiring multi-step mathematical calculations contextual to the problem. To effectively tackle the final question, our approach involves decomposing it into a sequential series of smaller sub-questions, allowing for a structured solution process.

**Hanoi Tower:** The Hanoi Tower problem (Gerety & Cull, 1986), a classic puzzle involving three pegs and a set of discs of varying sizes, serves as a key component of our experimental setup. The challenge requires moving the entire stack of discs from one peg to another, obeying the rules that only one disc can be moved at a time, and no disc may be placed on top of a smaller one. This task, structured around sequential and strategic disc placement, tests the model's ability to plan and execute a series of actions based on simple yet strict rules.

## 4.2 DATASET AND BASEMODEL

**Dataset:** The datasets we used are the open source datasets Blocksworld (Valmeekam et al., 2024), GSM8K (Cobbe et al., 2021a), AQUA (Ling et al., 2017), QASC (Khot et al., 2020), and our own production of Hanoi Tower. where the experiments for AQUA and QASC are in the Table 4.

**Basemodel:** The pre-trained models used in our study include: LLAMA-2-7B (Touvron et al., 2023), Phi-2-7B (Li et al., 2023b), Mistral-7B (Jiang et al., 2023) and Mixtral-8x7B (Jiang et al., 2024), Qwen1.5-7B (Bai et al., 2023), TAIDE-LX-7B[1], Mpt-7B (Team et al., 2023), Baichuan2-7B (Yang et al., 2023a),The model test results not mentioned in the main text will be supplemented in the Table 4 and 5.

## 4.3 BENCHMARK

**Train Parameter:** In this paper, we primarily utilize the 7B models for training on a single NVIDIA A100 GPU and models are loaded in 4-bit.

**RAP:** A technique that employs Monte Carlo Tree Search (MCTS) for exploration (Hao et al., 2023). RAP transforms LLMs into both reasoning agents and world models, utilizing MCTS for strategic exploration and decision-making. This approach significantly enhances the LLM's ability to generate action plans and solve mathematical and logical problems, outperforming traditional methods and establishing new benchmarks in LLM's capabilities.

**CoT:** A technique having enhanced the reasoning capabilities (Wei et al., 2022b) of LLMs. By providing models with intermediate reasoning steps as examples, CoT demonstrates notable improvements across various complex reasoning tasks, including arithmetic, commonsense, and symbolic reasoning. CoT requires the model to generate a reasoning chain to improve the reasoning ability. We used all basemodels to carry out CoT in the experiment.

**RoT:** A framework (Hui et al., 2024) to enhance the performance of tree-search-based prompting methods used in LLMs. This innovative approach leverages guidelines derived from past tree search experiences, allowing LLMs to avoid repeating errors and significantly improving their reasoning and planning capabilities across various tasks. We not only used the same basemodel as the original RoT, but also introduced other 7B models as a comparison.

## 4.4 RESULTS

**Blocksworld:** We conducted ablation experiments on the Blocksworld dataset. Our methodology, detailed in Section 3, particularly focuses on scenarios with more than 6 steps. As is shown in

---

[1]http://taide.tw

Table 2 and Table 5, for tasks up to 6 steps, results with our 7B models closely matched those with the benchmark's 70B models, suggesting robust inference capabilities even with reduced model size. For more complex tasks of 8 steps or more, DES improved its success rates by breaking down tasks into simpler segments, though it slightly lagged behind in performance compared to shorter tasks. This approach underlines the potential of our modified strategies in handling varying task complexities. By comparison, our CRE method not only outperforms benchmarks in terms of success rates on the 7B scale, but also achieves a higher success rate than the 70B+RAP method using the 7B model. For the arithmetic cases that use the full CreDes architecture, CreDes helps to improve the performance of the LLMs for long-range reasoning tasks.

**GSM8K:** We further independently verified the capabilities of CRE based on the GSM8K dataset without introducing DES, to confirm that it helps to enhance the inference capabilities of large models. We found that our CRE is superior to the baseline methods RAP, RoT, and CoT, further demonstrating that completing multi-step reasoning in one go has more advantages than completing multiple single-step reasoning. See Table 1. This example shows that CRE can not only help LLM solve highly structured problems, such as Blocksworld, but also has the ability to assist in solving some abstract mathematical problems.

**Hanoi Tower:** Unlike the Blocksworld case, the longest reasoning steps for the Hanoi Tower have a fixed quantitative relationship with the number of rods and disks. Therefore, when training the model, we used combinations within 7 steps, i.e., 3 rods and 3 disks. For evaluation, we used problems within 15 steps, i.e., combinations of 3 rods and 4 disks, to test the reasoning ability. From this perspective, our reasoning process is based on a zero-shot setting. Due to the time complexity of the search-based method for long-range reasoning, we did not conduct experiments for too many reasoning steps, and its success rate can be recorded as '-.' As Table 3 shows, CreDes performed best among all the models. By comparing the Hanoi Tower scenario with the Blocksworld scenario, we find that the success rate under Hanoi Tower is lower than that of Blocksworld, and that the reasoning ability of the 7B+CRE group is slightly lower than that of the 70B+RAP group. We believe that this phenomenon occurs because Hanoi Tower has a stricter stacking order qualification relative to Blocksworld, and some of the intermediate steps may not hold at all, see Fig. 2. From the results, the complexity of the Hanoi Tower problem is higher than that of Blocksworld.

**Time Efficiency:** Using the CRE and DES architecture has significantly shortened the time to complete long-range reasoning tasks compared to benchmarks, as is shown in Fig.5. This is because CreDes can perform simultaneous multi-step reasoning, which is more efficient than other methods that generate answers multiple times and then cascade them together, which is more evident in longer-range reasoning.

There is not much difference between the experimental results under 13B and 7B, and the difference can be regarded as a random error generated by different training. From the performance comparison between the 70B model and the 7B model under the RAP method and the 70B model, the performance of the 70B model will be relatively improved. However, considering inference speed, the 70B model is much slower than the 7B, and it needs to be loaded with a certain amount of quantization, and the performance loss is equally present.

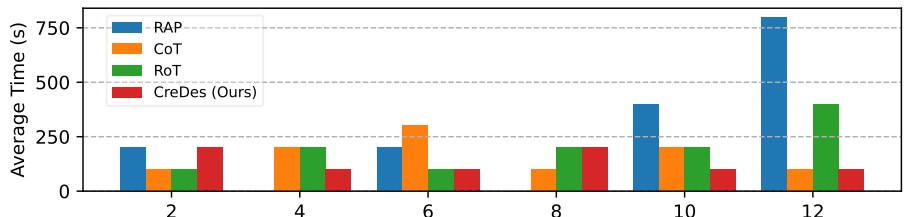

Figure 5: Improvement in reasoning speed for long-range tasks (based on a single A100 GPU).

## 4.5 DISCUSSION

This study introduced the CreDes framework, which combines CRE and DES to improve LLMs' ability to handle long-range reasoning tasks. CRE enhances robust causal relationships between

Table 2: Succcess Rate under Blocksworld

| Model | 2-step | 4-step | 6-step | 8-step | 10-step | 12-step |
|---|---|---|---|---|---|---|
| Llama-2-70B + RAP | 0.67 | 0.76 | 0.74 | 0.48 | 0.17 | 0.09 |
| Llama-2-7B + RAP | 0.39 | 0.41 | 0.37 | 0.11 | 0.00 | 0.00 |
| Llama-2-7B + CoT | 0.50 | 0.63 | 0.40 | 0.27 | 0.07 | 0.00 |
| Llama-2-7B + RoT | 0.52 | 0.67 | 0.27 | 0.06 | 0.00 | 0.00 |
| Llama-2-7B + CRE | **0.95** | **0.80** | **0.76** | 0.22 | 0.09 | 0.00 |
| Llama-2-7B + CreDes | - | - | - | **0.68** | **0.51** | **0.34** |
| Llama-2-13B + RAP | 0.44 | 0.42 | 0.38 | 0.11 | 0.00 | 0.00 |
| Llama-2-13B + CoT | 0.51 | 0.63 | 0.39 | 0.29 | 0.07 | 0.00 |
| Llama-2-13B + RoT | 0.49 | 0.70 | 0.30 | 0.07 | 0.00 | 0.00 |
| Llama-2-13B + CRE | **0.95** | **0.82** | **0.74** | 0.25 | 0.07 | 0.00 |
| Llama-2-13B + CreDes | - | - | - | **0.65** | **0.49** | **0.37** |
| Phi-2-7B + RAP | 0.40 | 0.44 | 0.33 | 0.00 | 0.00 | 0.00 |
| Phi-2-7B + CoT | 0.43 | 0.05 | 0.01 | 0.00 | 0.00 | - |
| Phi-2-7B + RoT | 0.54 | 0.16 | 0.01 | 0.01 | 0.00 | - |
| Phi-2-7B + CRE | **0.91** | **0.86** | **0.79** | 0.19 | 0.05 | 0.00 |
| Phi-2-7B + CreDes | - | - | - | **0.46** | **0.31** | **0.19** |
| Mistral-7B + RAP | 0.49 | 0.41 | 0.35 | 0.07 | 0.00 | 0.00 |
| Mistral-7B + CoT | 0.84 | 0.41 | 0.24 | 0.05 | 0.08 | - |
| Mistral-7B + RoT | 0.81 | 0.49 | 0.21 | 0.10 | 0.12 | - |
| Mistral-7B + CRE | **0.97** | **0.94** | **0.82** | 0.24 | 0.12 | 0.03 |
| Mistral-7B + CreDes | - | - | - | **0.54** | **0.37** | **0.21** |
| Mixtral-8x7B + RAP | 0.49 | 0.44 | 0.35 | 0.15 | 0.04 | 0.00 |
| Mixtral-8x7B + CoT | 0.81 | 0.63 | 0.55 | 0.18 | 0.20 | - |
| Mixtral-8x7B + RoT | 0.87 | 0.71 | 0.55 | 0.29 | 0.27 | - |
| Mixtral-8x7B + CRE | **0.99** | **0.97** | **0.93** | 0.34 | 0.22 | 0.13 |
| Mixtral-8x7B + CreDes | - | - | - | **0.75** | **0.57** | **0.40** |

reasoning steps, and DES can lower the complexity of long-range reasoning by using a bidirectional search approach. Our experiments, particularly in the Blocksworld and Hanoi Tower scenarios, demonstrated significant improvements in accuracy and efficiency over existing methods, implying that CreDes can effectively address the problem of causal hallucinations and huge search spaces.

## 4.6 LIMITATION

In scenarios with strict order of precedence, such as the Hanoi Tower, the accuracy is significantly lower compared to tasks like Blocksworld. The DES approach, while effective for moderate-length tasks, struggles with very long reasoning steps, leading to a decline in performance. Additionally, maintaining causal logic through CRE and DES introduces computational overhead, which may limit the framework's scalability and applicability in real-world scenarios with limited resources. Finally, our approach pays insufficient attention to the sequential ordering of steps, and the ATE can only determine whether the causal logic makes sense, rather than recognizing, for example, the assumption encountered in the Hanoi Tower problem that the larger disk must be placed under the smaller disk.

## 5 CONCLUSION

By integrating CRE and DES, the CreDes framework has significantly advanced LLMs' capabilities in long-range reasoning tasks. This combined approach enhances the accuracy and efficiency of multi-step reasoning and maintains the problem-solving and reasoning abilities of pre-trained models across different tasks. Future work will focus on refining the framework to improve scalability and efficiency in various complex problem-solving scenarios.

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

# A APPENDIX

## A.1 SUCCCESS RATE UNDER HANOI TOWER

Due to space constraints in the main text, we present the data from the Hanoi Tower experimental group here.

Table 3: Succcess Rate under Hanoi Tower

| Model | 3-step | 5-step | 7-step | 9-step | 11-step | 13-step |
|---|---|---|---|---|---|---|
| Llama-2-70B + RAP | 0.57 | 0.42 | 0.22 | 0.07 | - | - |
| Llama-2-7B + RAP | 0.29 | 0.21 | 0.11 | 0.00 | - | - |
| Llama-2-7B + CoT | 0.34 | 0.23 | 0.10 | 0.02 | 0.00 | 0.00 |
| Llama-2-7B + RoT | 0.41 | 0.27 | 0.13 | 0.04 | - | - |
| Llama-2-7B + CRE | **0.45** | **0.39** | **0.24** | 0.12 | 0.01 | 0.00 |
| Llama-2-7B + CreDes | - | - | - | **0.27** | **0.14** | **0.07** |
| Llama-2-13B + RAP | 0.30 | 0.20 | 0.12 | 0.00 | - | - |
| Llama-2-13B + CoT | 0.33 | 0.24 | 0.09 | 0.03 | 0.00 | 0.00 |
| Llama-2-13B + RoT | 0.44 | 0.30 | 0.12 | 0.03 | - | - |
| Llama-2-13B + CRE | **0.42** | **0.38** | **0.27** | 0.10 | 0.01 | 0.00 |
| Llama-2-13B + CreDes | - | - | - | **0.34** | **0.15** | **0.07** |
| Phi-2-7B + RAP | 0.27 | 0.21 | 0.14 | 0.01 | - | - |
| Phi-2-7B + CoT | 0.33 | 0.22 | 0.10 | 0.02 | 0.00 | 0.00 |
| Phi-2-7B + RoT | 0.24 | 0.12 | 0.02 | 0.00 | - | - |
| Phi-2-7B + CRE | **0.40** | **0.25** | **0.17** | 0.03 | 0.00 | 0.00 |
| Phi-2-7B + CreDes | - | - | - | **0.33** | **0.20** | **0.09** |
| Mistral-7B + RAP | 0.34 | 0.25 | 0.14 | 0.04 | - | - |
| Mistral-7B + CoT | 0.40 | 0.32 | 0.21 | 0.09 | 0.00 | 0.00 |
| Mistral-7B + RoT | 0.35 | 0.22 | 0.17 | 0.02 | - | - |
| Mistral-7B + CRE | **0.49** | **0.37** | **0.26** | 0.15 | 0.03 | 0.00 |
| Mistral-7B + CreDes | - | - | - | **0.37** | **0.19** | **0.11** |
| Mixtral-8x7B + RAP | 0.40 | 0.24 | 0.15 | 0.06 | - | - |
| Mixtral-8x7B + CoT | 0.45 | 0.27 | 0.14 | 0.02 | 0.00 | 0.00 |
| Mixtral-8x7B + RoT | 0.37 | 0.22 | 0.10 | 0.00 | - | - |
| Mixtral-8x7B + CRE | **0.50** | **0.35** | **0.22** | 0.11 | 0.01 | 0.00 |
| Mixtral-8x7B + CreDes | - | - | - | **0.42** | **0.25** | **0.12** |

## A.2 VALIDATION RESULTS OF MODEL'S INHERENT CAPABILITIES

To verify the success rate of our CRE method on other baseline tasks, we designed a control experiment to ensure that our approach does not impair the model's inherent problem-solving and reasoning abilities. Since DES is specifically designed for Blocksworld, a task with longer reasoning steps, the control experiments listed do not involve such lengthy reasoning steps; therefore, DES's performance is not tested in this section. The experimental results indicate that the CRE method can, to some extent, enhance the model's problem-solving capabilities on other baseline tasks without causing any reduction in performance. See Table 4.

## A.3 A NOTE ON THE HANOI TOWER DATASET

We generated and produced the Hanoi Tower dataset in the paper. The production method is to randomly generate several states conforming to the placement rules of the Hanoi Tower based on a given number of rods and disks, e.g., three rods and three disks, and randomly select one of these states as the starting and target states for a single sample. For a single sample, the classical partition algorithm is used to derive the pathway, and according to the length of the pathway, the sample is

Table 4: Results of model's inherent capabilities

| Model | AQUA | QASC |
|---|---|---|
| Llama-2-7B | 0.25 | 0.17 |
| Llama-2-7B + CRE | **0.74** | **0.62** |
| Baichuan-7B | 0.31 | 0.07 |
| Baichuan-7B + CRE | **0.85** | **0.31** |
| Mpt-7B | 0.11 | 0.05 |
| Mpt-7B + CRE | **0.65** | **0.27** |
| TAIDE-LX-7B | 0.27 | 0.21 |
| TAIDE-LX-7B + CRE | **0.89** | **0.72** |
| Qwen1.5-7B | 0.57 | 0.09 |
| Qwen1.5-7B + CRE | **0.75** | **0.37** |

categorized into different number of steps groups, e.g., 3-steps, 5-steps, 7-steps, and so on. An odd number is chosen for the allocation because the most complex solving step of Hanoi Tower in the case of three rods and $n$ disks is $2^n - 1$ steps. We generated the dataset Hanoi Tower using exactly the same storage format and Prompt structure as Blocksworld and GSM8K.

## A.4 PROMPT TEMPLATES USED DURING TRAINING AND TESTING OF CRE

**Prompt 1** Prompt Templates Used During **Training**

1: **Input:** Initial State || Goal State **####** Pathway
2: **Output: ####** Pathway
3: **Pathway:** <Step1><Step2><Step3><step4>

**Prompt 2** Prompt Templates Used During **Testing**

1: **Input:** Initial State || Goal State
2: **Output: ####** Pathway
3: **Pathway:** <Step1><Step2><Step3><step4>

## A.5 FULL EXPERIMENTAL RESULTS UNDER THE BLOCKSWORLD DATASET

## A.6 EXAMPLE: ERROR ANALYSIS

**Initial State:**

The blue block is clear, the orange block is clear, the hand is empty, the blue block is on top of the yellow block, the orange block is on top of the red block, the red block is on the table and the yellow block is on the table.

**Goal State:**

The orange block is on top of the yellow block.

**Expected Output:**

<unstack the blue block from on top of the yellow block>
<put down the blue block>
<pick up the orange block>
<stack the orange block on top of the yellow block>

Table 5: Succcess Rate under Blocksworld (Cont'd Table)

| Model | 2-step | 4-step | 6-step | 8-step | 10-step | 12-step |
|---|---|---|---|---|---|---|
| Baichuan-7B + RAP | 0.61 | 0.72 | 0.70 | 0.43 | 0.09 | 0.01 |
| Baichuan-7B + CRE | **0.93** | **0.74** | **0.71** | 0.25 | 0.05 | 0.00 |
| Baichuan-7B + CreDes | - | - | - | **0.63** | **0.47** | **0.29** |
| Mpt-7B + RAP | 0.25 | 0.06 | 0.00 | 0.00 | 0.00 | 0.00 |
| Mpt-7B + CRE | 0.32 | 0.11 | 0.04 | 0.00 | 0.00 | 0.00 |
| Mpt-7B + CreDes | - | - | - | 0.05 | 0.00 | 0.00 |
| TAIDE-LX-7B + RAP | 0.62 | 0.67 | 0.65 | 0.52 | 0.07 | 0.00 |
| TAIDE-LX-7B + CRE | **0.99** | **0.89** | **0.81** | 0.34 | 0.04 | 0.00 |
| TAIDE-LX-7B + CreDes | - | - | - | **0.70** | **0.54** | **0.35** |
| Qwen1.5-7B + RAP | 0.57 | 0.64 | 0.61 | 0.28 | 0.02 | 0.00 |
| Qwen1.5-7B + CRE | **0.92** | **0.77** | **0.73** | 0.34 | 0.08 | 0.02 |
| Qwen1.5-7B + CreDes | - | - | - | **0.61** | **0.46** | **0.36** |

**Actual Output:**

**RAP:**

```
A structure that unfolds in a tree shape:

First Layer:
<unstack the blue block from on top of the yellow block>

Second Layer:
<put down the blue block>
or <put down the yellow block> (hallucination)

Third Layer:
<unstack the orange block from on top of the red block> (hallucination)
or <unstack the orange block from on top of the yellow block> (hallucination)
or <unstack the orange block from on top of the blue block> (hallucination)

Fourth Layer:
<stack the orange block on top of the yellow block>
or <stack the orange block on top of the blue block> (pruned)
```

**CoT:**

```
The logic of CoT reasoning output is to solve complex problems by
step-by-step reasoning and refining intermediate steps, ensuring
the accuracy and reliability of the final answer.

First Input: Initial State
First Output:
<unstack the blue block from on top of the yellow block>
<put down the blue block>

Second Input: Initial State + First Output:
Second Output:
<pick up the blue block> (hallucination)
<stack the orange block on top of the blue block>
```

**CRE (Ours):**

```
Model one-time output of the whole process:
<unstack the blue block>
<put down the blue block>
<pick up the orange block>
<stack the orange block>

It should be clarified that CRE's mistake lies in the possibility
of incomplete answers as mentioned above.
```

### A.7 ASSUMPTIONS

**Assumption 1.** The experimental observation outcomes for any sample do not vary with the treatment assigned to other samples, and, for each sample, there are no different forms or versions of each treatment level, which lead to different experimental observation outcomes.

**Assumption 2.** Given the background variable $X$, treatment assignment $W$ is independent of the potential outcomes, i.e., $W \perp\!\!\!\perp Y(W = 0), Y(W = 1) \mid X$.

**Assumption 3.** For any value of $X$, treatment assignment is not deterministic:

$$P(W = w \mid X = x) > 0, \quad \forall w \text{ and } x. \tag{7}$$

With these assumptions, the relationship between the observed outcome and the potential outcome can be rewritten as:

$$\mathbb{E}[Y(W = w) \mid X = x] = \mathbb{E}[Y(W = w) \mid W = w, X = x]$$
$$= \mathbb{E}[Y^F \mid W = w, X = x], \tag{8}$$

where $Y^F$ is the random variable of the observed outcome, and $Y(W = w)$ is the random variable of the potential outcome of treatment $w$.

### A.8 TREATMENT EFFECT

With the above Assumptions, we can rewrite the Treatment Effect defined as follows:

$$\text{ITE}_i = W_i Y_i^F - W_i Y_i^{CF} + (1 - W_i) Y_i^{CF} - (1 - W_i) Y_i^F \tag{9}$$

$$\text{ATE} = \mathbb{E}_X \left[ \mathbb{E}[Y^F \mid W = 1, X = x] - \mathbb{E}[Y^F \mid W = 0, X = x] \right]$$

$$= \frac{1}{N} \sum_i \left( Y_i(W = 1) - Y_i(W = 0) \right) = \frac{1}{N} \sum_i \text{ITE}_i \tag{10}$$

$$= \mathbb{E}(Y \mid do(X)) - \mathbb{E}(Y) = \mathbb{E}[Y_1 - Y_0]$$

where $Y_i(W = 1)$ and $Y_i(W = 0)$ are the potential treated/control outcomes of sample $i$, $N$ is the total number of samples in the whole dataset. The second line in the ATE is the empirical estimation. Empirically, the ATE can be estimated as the average of the ITE across the entire dataset. In equation 10, $do(\cdot)$ refers to Do-calculus**?**, which denotes an external intervention on the value of $X$ without affecting the actual state of $Y$.

### A.9 ADDITIONAL DETAILS

**Dataset Validity and Construction:** The Hanoi Tower dataset is more complex than Blocksworld, involving a judgment on stacking order. Errors arise if the stacking order is violated, making the task harder. The dataset's size matches Blocksworld, with all steps being odd numbers based on the minimum steps required.

**Computational Efficiency and Scalability:** The 7B models fit within a single A100 GPU which is mentioned in paper. The 13B models have similar time requirements, as quantization isn't needed. However, 70B models experience significant speed drops, likely due to quantization and their size.

**Theoretical Background and Practical Considerations:** We perform numerous output trials to calibrate the model during the training process. From our experimental results, these output samples demonstrate a variety of possibilities, such as:

Type 1: Certain samples are challenging to answer correctly, regardless of training, resulting in near-random correct/incorrect states.

Type 2: There is a positive correlation between epoch count and correct answer frequency for some samples, significantly when aided by standard training techniques like RAP and CoT.

Type 3: Some samples can be answered correctly with minimal training, showing no correlation between epoch count and correct answer frequency.

As Blocksworld researchers, the current goal is to maximize the correct rate of Type 1 samples, effectively converting more Type 1 samples into Type 2.

**Experimental Comparison about Embodied Intelligence:** Many real-world reasoning and long-range sequence decomposition tasks fundamentally adhere to the same paradigm as the one employed in our research. We recognize that there may be lingering concerns regarding the practical applicability of our approach in real-world scenarios. We offer the following clarifications to address these concerns, supported by practical examples. Established algorithms are widely used within the logistics industry in areas such as port container scheduling or the organization of goods in warehouse facilities. Our research, however, aims to augment these processes by leveraging Large Language Models (LLMs) to enhance reasoning capabilities within these contexts. The primary goal is to bridge the communication gap between human operators and algorithm engineers, allowing LLMs to facilitate more transparent and effective interactions. By understanding and interpreting human instructions, we hypothesize that LLMs can dynamically adjust their outputs, thereby improving collaboration between human operators and algorithmic systems.

Although our approach has yet to be validated with real-world data, we emphasize that the nature of many real-world reasoning or long-range sorting tasks closely mirrors the experimental paradigm used in our study.

To further clarify, consider the example of warehouse item arrangement. This process involves organizing goods according to criteria such as size, weight, or frequency of access. While this is a complex, unified task, it can be decomposed into smaller, interrelated sub-tasks. For instance, the initial task may be categorizing items by size, arranging them within sections based on weight, and finally, positioning them according to access frequency. Each sub-task depends on the previous one, forming a continuous sequence of actions that ultimately leads to completing the overall task.

It is worth noting that several related studies do not explicitly connect their experiments to real-world applications. However, the scope of our experiments is comparable to that of other works in the field. In particular, other research has adopted similar test scenarios and datasets, further reinforcing our confidence in the robustness of our experiments.

