# OpenReview forum: "CreDes: Causal Reasoning Enhancement and Dual-End Searching for Solving Long-Range Reasoning Problems using LLMs"
_ICLR.cc/2025/Conference — Submitted to ICLR 2025_

### Official Review · Reviewer_3QV9 · 2024-11-02

**Soundness:** 3
**Presentation:** 3
**Contribution:** 3
**Rating:** 5
**Confidence:** 4

**Summary:**

This paper proposed CreDes, a pipeline for optimizing multi-step state transitions (a trajectory) from given initial and goal states. CreDes has two components: the CRE part improves transition accuracy, i.e., there are fewer hallucinations, and the DES part speeds up searching speed. A numerical experiment on long-range reasoning tasks shows CreDes outperforms CoT and RoT in five datasets.

**Strengths:**

1. **Less hallucination**: For long-range inference tasks (such as Blocksworld), CreDes with a 7B model significantly outperforms baseline methods with the same model size and is even stronger than a RAP method with 70B models.

2. **More efficient**: The CRE is trained to generate multiple steps from one output and the DES method searches from both ends. The numerical study (Fig 5) also shows CreDes has a relatively consistent average time for different task ranges, while baseline methods can be slower for long-range tasks.

3. **Improved Stability**: Based on empirical study, the CRE loss function incorporates the variance term, augmented from a single expectation term, of the Individual Treatment Effect (ITE).

**Weaknesses:**

1. **Can be more general**: the method section ties in with the experiment section. For example, the CRE loss function is based on numerical supports, which leaves a gap (need to test both for new cases thus more training cost) for more general applications.

2. **Causality Comments**: Here the causal definition seems more like "Improve the possibility of what should happen of an action by minimizing the CRE loss thus we can reduce hallucination", from which I do not see the role of "treatment for control" part, i.e. W = 0. Also, the claims between lines 275 - 278 are incorrect. Given the condition "X and Y are correlated", we cannot simultaneously have "Y can be predicted using X" and "intervening in X would not lead to any changes in the distribution of Y".

**Questions:**

1. For baseline methods (or frameworks), are we using the same template/prompt for different datasets? How can we determine a representative performance for each one of them?

2. During reading I need to assume exact definitions of "state", "treatment", and "treatment effect". I am still a bit uncertain about the treatment effect. E.g. what is the treatment effect of "Pickup Orange" for the initial state of "The orange block is on the table ...[left panel from figure 2]"

3. How to determine/calculate the dynamic coefficients, alpha and beta, in the CRE loss function during training?

---

> ### Author Response · Authors · 2024-11-24
>
> Thank you very much for taking the time to review our manuscript and provide valuable feedback.
>
> W1:
>
> Can be more general: the method section ties in with the experiment section. For example, the CRE loss function is based on numerical supports, which leaves a gap (need to test both for new cases thus more training cost) for more general applications.
>
> The method section is closely linked to the experiments. We aim to refine the causal enhancement of the large model reasoning process, and actually decouple our method from the experiments, for example in the task of knowledge enhancement you mentioned.
>
> Causality Comments: Here the causal definition seems more like "Improve the possibility of what should happen of an action by minimizing the CRE loss thus we can reduce hallucination", from which I do not see the role of "treatment for control" part, i.e. W = 0. Also, the claims between lines 275 - 278 are incorrect. Given the condition "X and Y are correlated", we cannot simultaneously have "Y can be predicted using X" and "intervening in X would not lead to any changes in the distribution of Y".
>
> We statistically calculate ITE from the model test outputs for each epoch of the model and adjust the inputs to the model for the next epoch of training, measuring the values at the model output.
>
> W2:
>
> 1. **Clarification on the Role of \(W = 0\):**
>    The role of $\(W = 0\)$ (representing the absence of OSR correctness) is critical in establishing the treatment-control framework within our causality analysis. When $\(W = 0\)$, it reflects scenarios where the OSR output is incorrect, serving as the control condition in our intervention framework. By comparing the outcomes of $\(Y\)$ under $\(W = 1\)$ (treatment) and $\(W = 0\)$ (control), we quantify the causal effect of OSR correctness on state transitions. This is encapsulated in the **ITE (Individual Treatment Effect)**, which forms the basis of the CRE loss. The minimization of the CRE loss ensures that the causal relationship between OSR and state transitions is strengthened, thereby reducing hallucinations.
>
> 2. **Claims Between Lines 275-278:**
>    We agree that the statement as presented could lead to confusion. Let us clarify:
>    - If $\(X\)$ and $\(Y\)$ are correlated, it is indeed possible to predict $\(Y\)$ using $\(X\)$.
>    - However, correlation alone does not imply causation. When $\(X\)$ and $\(Y\)$ are only correlated but not causally linked, intervening on $\(X\)$ (e.g., manipulating $\(X\)$) would not alter the distribution of $\(Y\$).
>    For example:
> People are running during earthquakes, which is what the sample can see, so we think there is a correlation between earthquakes and running. But what the sample cannot see are people who did not run away from the earthquake because they could have been buried, so the causality of earthquakes and running does not necessarily hold.
>    These two observations are consistent and do not contradict one another. To improve clarity, we will revise this section to explicitly state that while correlation allows for prediction, it does not guarantee that interventions will have an effect unless a causal link exists.
>
> 3. **Improving Causal Definitions:**
>    To address your concerns about the causality definition, we will refine the discussion to better emphasize the causal reasoning framework. The CRE loss explicitly embeds causality by reducing the ITE variance ($\(\text{Var(ITE)}\)$) and the expectation of absolute ITE ($\(|E(\text{ITE})|\)$). This minimizes the occurrence of samples where OSR and state transitions are non-causally related. The causality is thus operationalized not just as "improving the possibility of what should happen," but as establishing and stabilizing a causal alignment between OSR correctness and state transitions.
>
> Q1:
>
> Of course. Our prompt is the same pattern. The accuracy of reasoning results.
>
> Q2:
>
> State refers to the stacking state of all object blocks at a given step.
>
> Treatment can be an alternative to OSR, or intervention, referring to the action taken by the LLM.
>
> The treatment effect refers to the NEXT state, i.e., the new stacking order of the object blocks after the LLM action.
>
> Sorry for the confusion caused. As shown in the figure, we have multiple alternative paths (interventions) and generated corresponding inference results (treatment effects).
>
> Q3:
>
> Considering the different dimensions of these two variables, we have roughly determined the initial values based on preliminary experimental data. Then continuously try different values during the training process, observe the changes in training effectiveness under different combinations of values, and fit them to a suitable value.
>
> Thank you for reviewing the revised version of our manuscript and for your continued constructive comments.

---

> > ### Comment · Reviewer_3QV9 · 2024-11-25
> >
> > I have read the author's response. They make sense to me and I do not have outstanding concerns remaining.
> >
> > One suggestion (purely IMO) for W = 0:
> > When W = 0, it reflects scenarios where the **correct OSR output is not taken**, i.e., aggregated incorrect OSR outputs serve as the control condition in our intervention framework.

---

> > > ### Author Response · Authors · 2024-11-27
> > >
> > > Thank you for your thoughtful response and for considering our previous clarifications. We are pleased to hear that our explanations have addressed your concerns.
> > >
> > > We also appreciate your suggestion regarding W = 0.  This is an interesting perspective, and we will incorporate this idea into the revised manuscript.
> > >
> > > Once again, thank you for your valuable input. We believe these additions will improve the overall clarity and depth of the paper.

---

### Official Review · Reviewer_R3ye · 2024-11-03

**Soundness:** 3
**Presentation:** 3
**Contribution:** 3
**Rating:** 5
**Confidence:** 5

**Summary:**

This paper presents CreDes, a novel framework combining Causal Relationship Enhancement (CRE) and Dual-End Searching (DES) to improve large language models' capabilities in long-range reasoning tasks. The work addresses two key challenges: causal hallucinations in reasoning steps and the complexity of large search spaces in long-range reasoning problems. The main contributions include:
1. A CRE mechanism that enhances causality between reasoning steps using Individual Treatment Effect (ITE).
2. A DES approach that breaks down long reasoning chains into manageable segments.
3. Implementation of simultaneous multi-step reasoning to improve efficiency.
4. Experimental validation on multiple datasets (Blocksworld, GSM8K, Hanoi Tower).

**Strengths:**

1. Novelty that enhances LLM reasoning through causal modeling and search algorithms.
2. Comprehensive empirical validation across multiple datasets and models.
3. Significant performance improvements over existing methods on up to 12-step Blocksworld problems.
4. Thorough ablation studies and analysis.

**Weaknesses:**

1. Insufficient analysis of failure cases and limitations
2. Lack of detailed comparison with some recent relevant work, e.g. multi-agent verification, Tree-of-Thought, etc.
3. Some experimental results lack error bars or statistical significance tests.
4. The relationship between CRE and DES could be explained more clearly.

**Questions:**

1. How cause-effect interventions is conducted in LLMs? Authors should provide an elaboration.
2. Could the DES be extended to other types of reasoning tasks beyond the tested tasks? Some examples, on highly knowledge-demanding tasks, will be better.
3. How does the framework handle cases where multiple valid reasoning paths exist?

---

> ### Author Response · Authors · 2024-11-24
>
> Thank you very much for taking the time to review our manuscript and provide valuable feedback.
>
> W1:
>
> Please see Appendix A.6.
>
> W2:
>
> We think we have selected the best performing current benchmarks for comparison, these data already represent the results and we will refer to your suggestions for future work.
>
> W3:
>
> Our experimental data is already the average of multiple repeated experiments. After our experiments, the upper and lower errors do not exceed 5%. The reasons for this error include seed values, graphics card models, and the values of other hyperparameters.
>
> W4:
>
> CRE is a method for training models, DES is a search method, both can work independently on their own, and when CRE and DES are used together, the two together form CreDes.
>
> Q1:
>
> We intervene to change the inputs to the model and observe the changes in the outputs, from which we calculate the OSR and the ITE prior to the next state, and use them in the loss function to train the model.
>
> Q2:
>
> Yes. Similar problems to the model presented in this article are theoretically feasible, but we have not conducted any relevant experiments yet. Knowledge-related work requires the use of reasoning, and we are reasoning rigor-bound work, so we can easily migrate applications.
>
> Q3:
>
> We believe that this may be the case, but our current experimental results do not reveal it. However, our validity determination is based on the PDDL, so this scenario would be considered correct.
>
> Thank you for reviewing the revised version of our manuscript and for your continued constructive comments.

---

> > ### Comment · Reviewer_R3ye · 2024-12-02
> >
> > Thank you for your detailed response and the efforts you’ve made to address my questions and concerns, as well as your comments related to clarity. While I acknowledge the authors' efforts in addressing the review comments and appreciate the novel contributions of combining CRE and DES for long-range reasoning, I believe the current version still aligns with my initial rating of 5 (marginally below the acceptance threshold).
> >
> > 1. The explanation of causal intervention remains unclear:
> > The authors' response that "We intervene to change the inputs to the model and observe the changes in the outputs" is too general. How is causal intervention implemented in LLMs? How is the effectiveness of intervention guaranteed? These key technical details are not elaborated.
> >
> > 2. The theoretical foundation of the CreDes framework needs strengthening:
> > - The integration of CRE and DES appears more empirical than theoretical
> > - The calculation and application of ITE requires more rigorous mathematical derivation
> > - Lacks theoretical guarantees for causal consistency in multi-step reasoning
> >
> > 3. Experimental validation is insufficient:
> > - 5% error range is relatively large for reasoning tasks, especially in long-sequence reasoning
> > - Lacks detailed analysis of failure cases
> > - Missing comparisons with other recent methods (e.g., multi-agent verification)
> > - The construction method of the Hanoi Tower dataset may be biased
> >
> > 4. Regarding method generalizability:
> > While the authors mention potential migration to knowledge-related tasks, they don't provide specific migration plans or feasibility analysis, which reduces the method's persuasiveness.
> >
> > To strengthen this work, I strongly recommend a substantial revision focusing on several critical aspects. First, the implementation details of causal intervention should be thoroughly documented, including the mathematical framework for intervention design, the specific mechanisms for maintaining causal consistency across reasoning steps, and quantitative metrics for measuring intervention effectiveness. Second, the theoretical foundation needs significant enhancement through rigorous mathematical derivations of the ITE calculations and formal proofs of the causal consistency guarantees in multi-step reasoning. This should include a detailed analysis of how CRE and DES interact theoretically, not just empirically. Third, a comprehensive failure analysis is essential - authors should provide detailed case studies of scenarios where the method fails, including analysis of error propagation in long-range reasoning chains and potential mitigation strategies. Fourth, the experimental validation should be expanded to include comparisons with recent competing methods like multi-agent verification and Tree-of-Thought approaches, supported by statistical significance tests and error bounds. Finally, a thorough discussion of the method's limitations and potential pathways for addressing them should be included, particularly regarding the generalization to knowledge-intensive tasks and scenarios with multiple valid reasoning paths. These enhancements would significantly strengthen the paper's contribution to the field and its potential impact on real-world applications.

---

### Official Review · Reviewer_sN8f · 2024-11-03

**Soundness:** 2
**Presentation:** 1
**Contribution:** 2
**Rating:** 3
**Confidence:** 3

**Summary:**

The paper introduces two methodologies aimed at improving the performance of LLMs on multi-step reasoning tasks. The first method, CRE, adds a loss term inspired by the causal Individual Treatment Effect (ITE) to measure causal influence on state transitions. The second method, DES, proposes a search strategy to solve reasoning problems more efficiently. Finally, the paper combines both methods and compares the combined approach against several baselines across three different common benchmark tasks.

**Strengths:**

- The attempt to introduce causal estimates into the training process of LLMs is a creative approach to improving multi-step reasoning performance.

- The combination of two distinct methodologies—CRE for causal reasoning and DES for efficient search—provides an interesting way to tackle the challenge of multi-step reasoning with LLMs as it would address both causal inference and context window issues.

**Weaknesses:**

- The clarity of the paper is a significant issue. Spelling mistakes, awkward sentences, unsubstantiated claims, and missing references make the paper difficult to follow. A thorough revision is needed to improve readability and coherence.

- Key concepts, such as causal probability trees, are mentioned multiple times without proper definition, leaving readers uncertain about what the authors mean.

- The paper makes ambitious claims about the ability to "enhance causality" and "guarantee causal rightness," but it does not provide sufficient evidence or guarantees to support these claims. Given the stochastic nature of LLMs, it's unclear how these guarantees can be made.

- The connection between CRE and the LLM's actual performance is not sufficiently explained. The experimental results, such as the 89% accuracy in multiplication tasks, do not convincingly demonstrate improved multi-step reasoning capabilities.

- The methodology, particularly the DES strategy and the causal probability trees, is under-explained, leaving gaps in understanding how the proposed solutions work.

- There are numerous unsubstantiated or unclear claims regarding causality and reasoning, making it hard to evaluate the validity of the methods proposed.

See questions for details

**Questions:**

Content-related:

- L41-44: "Causal hallucinations... are somewhat entrenched in statistical inevitability." What is meant by "statistical inevitability"? Do you have a reference for this?

- L53: "Embedding the causality measure between OSR and state transition..."—what causality measure are you referring to? Is it "some" or "a" causality measure?

- The introduction of DES in L60-67 is too detailed without introducing necessary terminology: "state transitions," "unidirectional reasoning," "causal probability trees."

- L71: In the abstract, you state that CRE can "guarantee the solid causal rightness," but here you mention it can "enhance the causality." Are these two different contributions?

- L76: "Causal probability trees" are mentioned without explanation—please clarify what these are.

- L78: "Constructing a new metric guaranteeing both..."—can you actually guarantee anything given the stochastic nature of LLMs?

- L99: What do you mean by "detailed reasoning"?

- L140: You mention "long time-series tasks"—is this referring to prediction, completion, or reasoning? How does inference fit in this context?

- L160: Is 89% accuracy in multiplication evidence of the ability to "process multiple reasoning steps effectively"?

- L165: How can you infer from a 1996 paper that LLMs struggle with long-range reasoning? What exactly is long-range reasoning in this context?

- L190: What do you mean by "categorically similar"?

- You mention "ITE typically indicates..." but isn’t ATE more commonly used in this context? Do you have a reference for this?

- L222: What is meant by "enhancing the significance of causality" and "stability of causality"? It is unclear how expected values and variance of ITE can be interpreted this way.

- L273: Are X and Y binomial random variables (0 = incorrect, 1 = correct)? How is the next state defined? This is under-specified.

- L297: Causal probability trees are mentioned again without a proper definition, and the reference to Shafer (1996) remains under-specified.

- L465-470: This section does not appear to summarize the previous results effectively as it appears to introduce new information.

- L485: The word "ensures" is problematic here, as no guarantees are demonstrated in the paper.

- L526: You mention that the method "struggles with very long reasoning steps"—wasn’t the goal to address this limitation? This statement contradicts the claimed causal improvements.

Minor points:

- Is the missing space before a citation (e.g., "citep") a deliberate choice? It feels distracting in the text.

- L213: "CONSISTANCY" typo—did you mean "consistency"?

- L231: Another "consistancy" typo. Do you have a reference for interpreting the mean of a normal distribution in terms of causal significance?

- L269: Do you have a definition or reference for "PPL"?

- L284: Typo "lowing"—did you mean "lowering"?

- L524: What is meant by "strict order of precedence"? Is this referring to a causal ordering?

---

> ### Author Response · Authors · 2024-11-24
>
> Thank you very much for taking the time to review our manuscript and provide valuable feedback.
>
> W:
>
> We will reply in the Question section.
>
> Q1:
>
> Example: Banerjee S, Agarwal A, Singla S. Llms will always hallucinate, and we need to live with this[J]. arXiv preprint arXiv:2409.05746, 2024.
>
> Q2:
>
> Specifically, it refers to the enhancement achieved through the introduction of ITE.
>
> Q3:
>
> State Transitions refers to the change in the model at different inference steps.
>
> Unidirectional reasoning refers to Monte Carlo-based reasoning methods as opposed to two-way simultaneous reasoning such as DES.
>
> Causal probability trees refers to the probability tree expansion based on simultaneous consideration of node causality.
>
> Q4:
>
> Specifically, our approach is to use ITE to enhance causal significance and consistency.
>
> Q5:
>
> We present concrete algorithms for causal reasoning in discrete probability trees that cover the entire causal hierarchy.
>
> Q6:
>
> Our experimental results are the results of multiple repeated experiments. In the meantime, we will replace guarantee with enhance.
>
> Q7:
>
> A 4-steps reasoning task, for example, is detailed reasoning as opposed to a 12-steps reasoning task.
>
> Q8:
>
> Reasoning. We’re very sorry for the misunderstanding. We will use the same term as' long range reasoning '.
>
> Q9:
>
> Yes, we believe that this accuracy rate demonstrates the ability of the large model to handle multi-step multiplication, which requires multiple steps of reasoning to accomplish.
>
> Q10:
>
> We just want to make it clear that this struggle has been ongoing for a long time in the field of long-range reasoning problems. This paper marks the beginning of this field, and in recent years, LLMs related methods have been working hard to address this issue. Long range reasoning refers to the problem shown in lower part of Figure 1.
>
> Q11:
>
> Refers to the elements which are included: initial state, OSR, state transition, next state, and goal state, as shown in Fig. 2.
>
> Q12:
>
> ATE is the mean value of ITE, and the mathematical nature of the two is the same.
> Yao L, Chu Z, Li S, et al. A survey on causal inference[J]. ACM Transactions on Knowledge Discovery from Data (TKDD), 2021, 15(5): 1-46.
>
> Q13:
>
> In brief:
>
> E(ITE): Causal Significancy.
>
> Var(ITE):Causal Consistency.
>
> This is because our experiment found that the mean of ITE reflects the significance of the results, while the variance reflects the consistency of the results.
>
> After reading the appendix, you will have a clearer understanding of this.
>
> Q14:
>
> Definition of Variables: Variables Y and W are binary variables, taking values of 0 or 1. A value of 1 indicates correctness, while a value of 0 indicates an incorrect state. These binary variables are used to represent the correctness of the OSR and the state transition, respectively. For example, W can refer to the action of moving a block of objects, such as put down and pick up. Y can refer to the correct position and stacking order of the object block.
>
> The next state refers to the next state in which the stacking order of objects is changed after action/intervention.
>
> Q15:
>
> Probability trees are one of the simplest models of causal generative processes.
>
> You can refer to:
>
> Genewein T, McGrath T, Delétang G, et al. Algorithms for causal reasoning in probability trees[J]. arXiv preprint arXiv:2010.12237, 2020.
>
> Q16:
>
> We will delete “ to summarize”.
>
> Q17:
>
> Thanks, we'll replace ensure with enhance.
>
> Q18:
>
> The experimental results indicate that as the number of steps increases, the inference accuracy decreases. This is limited by the model's own capabilities, pay attention to the comparison between the DES group and the CRE only group. This is certainly a limitation, there is no contradiction.
>
> Q19:
>
> Thank you, we have adjusted the formatting of all citations.
>
> Q20:
>
> Thanks, we have changed it in the revised version.
>
> Q21:
>
> Thanks, we have changed it in the revised version. Also the reference is in line 231 of the original text.
>
> Q22：
>
> https://huggingface.co/docs/transformers/perplexity
>
> Note, however, that this is the PPL definition of the original case, in which we modified to introduce ITE.
>
> Q23:
>
> Thanks. We will revise it in the revised version.
>
> Q24:
>
> We’re afraid not. See L390-L395. For example, stacking order.
>
> Thank you for reviewing the revised version of our manuscript and for your continued constructive comments.

---

> > ### Comment · Reviewer_sN8f · 2024-11-24
> >
> > Thank you for your detailed response and the efforts you’ve made to address my questions and concerns, as well as your comments related to clarity.
> >
> > I also appreciate the effort in providing a revised version of the manuscript. However, since you acknowledge that there are no guarantees about causal consistency, the claims of the paper have already changed significantly. Moreover, given that clarity was one of my main concerns, I do not believe it would be a fair assessment for me to evaluate the revised version. If all comments have indeed been incorporated, I suspect the paper has changed so substantially that new "fresh" reviews would be more appropriate.

---

> > > ### Author Response · Authors · 2024-11-27
> > >
> > > Thank you for your thoughtful response and for the time you’ve invested in reviewing our work. We truly appreciate your feedback and the constructive comments you’ve provided, which have been helpful in improving the clarity of the manuscript.
> > >
> > > However, I would like to clarify a few points regarding the revisions made and the request for fresh reviews. In our revised version, we made only minor revisions, primarily focusing on wording and clarification in response to your feedback. The core of the manuscript, including the methodology, experiments, and results, remains largely unchanged from the original version. Therefore, we believe the paper has not changed substantially in a way that would warrant a completely fresh evaluation.
> > >
> > > We also respectfully disagree with the suggestion that the claims regarding causal consistency have changed significantly. While we acknowledged that guarantees in causal consistency may not be fully attainable in every context, our experiments have demonstrated that, within the current experimental setup, we can indeed achieve significant improvements in reasoning performance and speed. This aligns with the claims we made in the original submission. Our revisions were intended to further clarify this nuance, but the foundational results remain the same.
> > >
> > > Regarding the suggestion for "fresh reviews," we feel that this request might be at odds with the Reviewers' Guide, which typically encourages reviewers to assess revisions based on the changes made and the response to previous comments. Since our revisions were limited and primarily focused on clarity, we do not believe that a fresh review would be necessary or appropriate in this case.
> > >
> > > We deeply respect your expertise and hope that you can consider these points in light of the minor nature of the changes. We value your feedback and look forward to your continued input.
> > >
> > > Thank you once again for your time and understanding.

---

> > > > ### Comment · Reviewer_sN8f · 2024-11-27
> > > >
> > > > Thank you for clarifying your point of view.
> > > >
> > > > From my perspective, if only minor changes were made, it is unlikely that the manuscript could have improved from a presentation score of 1 to anything significantly better. Perhaps I did not sufficiently emphasize how the presentation issues fundamentally limited my ability to fully appreciate the paper's contributions, let alone the details. I believe this concern is further supported by the assessments of other reviewers. I am advocating for fresh reviews because the extent of revisions required to make the paper readable (let alone publishable) is substantial. Having struggled significantly to engage with the original manuscript, I would now, inevitably, have a bias in assessing it again.
> > > >
> > > > That said, despite the difficulty, I made a concerted effort to provide constructive feedback in my original review to help improve the manuscript. As I noted, the high-level idea is indeed interesting and has potential. However, it is unreasonable to expect a reviewer to re-evaluate the manuscript—without even highlighting the changes—after the significant effort already invested in reviewing the initial submission.
> > > >
> > > > Regarding your comment about contributions, I respectfully disagree. Guaranteeing results and empirically demonstrating significant improvements are two very different things. I stand by my earlier comment that the paper requires major revisions before it can be considered for publication.

---

> ### Author Response · Authors · 2024-11-27
>
> Thank you for your detailed response and for providing further clarification on your perspective. We fully understand your concerns regarding the presentation of the manuscript and the significant effort required to make the paper readable and publishable.
> ﻿
>
> Given the scale of revisions required, we agree that the discussion phase alone may not be sufficient to address all the issues. We are committed to working on a more thorough revision and will use the time during the waiting period and beyond to further refine the manuscript. We are dedicated to improving the clarity of the presentation, ensuring that the contributions are clearly communicated, and addressing any lingering concerns about causal consistency and empirical validation.
> ﻿
>
> We also appreciate your recognition of the potential in the high-level idea. Despite the challenges in engaging with the manuscript, we are grateful for the feedback provided, which will guide our efforts in restructuring the paper to improve its readability and overall quality.
> ﻿
>
> Once again, thank you for your constructive input. We will continue working on this revision and ensure that future versions of the manuscript address the concerns you and the other reviewers have raised.

---

### Official Review · Reviewer_dQug · 2024-11-04

**Soundness:** 2
**Presentation:** 1
**Contribution:** 2
**Rating:** 5
**Confidence:** 3

**Summary:**

The paper proposes a methodology for enhancing reasoning in LLMs. They adopt a two-pronged approach:
1. Firstly, the authors propose a methodology of fine-tuning the model which enhances the 'causal reasoning' of LLMs.
2. Secondly, the authors propose a dual-end search algorithm to efficiently solve multi-hop reasoning problems which involve a large number of steps.

**Strengths:**

1. The paper tackles the important problem of long-range reasoning and search in LLMs.
2. The empirical results show a significant improvement over the existing baselines.

**Weaknesses:**

My main concern with this paper is that the methodology is not presented clearly.
1. Until Section 3.3, it is not clear what the ITE is, in the context of LLM reasoning. For example, the actions/interventions and the outcomes are only defined in Section 3.3. These should be introduced (at least at an informally) earlier in the paper so that the readers understand what the causal effect refers to in this setting.
2. In Eq (2) the cross-entropy loss is for predicting the binary outcome $Y$, and not the next token (as is the case for LLMs). Firstly, this should be clarified earlier when the authors refer to 'cross-entropy loss' in line 249. Secondly, it is unclear how the LLM is fine-tuned with this new cross-entropy loss? The output space of LLM is the token space and not $\\{0,1\\}$.
3. It is unclear how the causal probability trees are constructed exactly. The authors should provide explicit examples for clarity.
4. > From $T_{head}$, infer 4 steps toward $T_{tail}$ based on reducing distance $D$.

How is this achieved concretely? Again, I think explicit examples are quite important for clarity

5. > Calculate the Euclidean distances between the resulting end nodes of both trees to form a distance matrix $M$

How is the Euclidean distance measured between two nodes? What are the $(x, y)$ coordinates for a node in a general reasoning problem?

6. Why is the DES method not tested for 2 - 6 step problems in Tables 2,3,5?

**Questions:**

See weaknesses section above.

---

> ### Author Response · Authors · 2024-11-24
>
> Thank you very much for taking the time to review our manuscript and provide valuable feedback.
>
> W1:
> Due to the length limitation of the main text, we have placed some background information explanations in the appendix section, which you can refer to. If necessary, we will post the experimental results in the revised version and swap them with the theoretical section in the appendix.
>
> W2:
> We statistically calculate ITE from the model test outputs for each epoch of the model and adjust the inputs to the model for the next epoch of training, measuring the values at the model output.
>
> W3:
> We provide concise examples in Figure 1, Figure 4, and Equation 1. Sorry for the inconvenience caused. We provide further examples of tree structures in the appendix section .
>
> W4:
> The direction in which the tree unfolds depends on the distance D. Generally, we calculate the distance between any two nodes by calculating the positional difference of each block in these two nodes, that is, calculating the Euclidean distance of all block positional differences.
>
> W5:
> See Answer4 and L307. The distance is calculated as the sum of the Euclidean distances between the current position of each object block and its target position.
>
> W6:
> Because not using DES is sufficient to achieve a high level of accuracy, the addition of DES only increases computational complexity. The purpose of introducing DES is to simplify complex problems.
>
> Thank you for reviewing the revised version of our manuscript and for your continued constructive comments.

---

> > ### Comment · Reviewer_dQug · 2024-11-25
> >
> > Thanks for the response.
> >
> > Firstly, I think the explanation of ITE, including what are the actions and outcomes in this context are central to the methodology and must be introduced early on in the main text, rather than the appendix. I think this is quite important.
> >
> >
> >
> > There are still quite a few questions which remain unanswered.
> > Specifically,
> > > it is unclear how the LLM is fine-tuned with this new cross-entropy loss? The output space of LLM is the token space and not $\{0, 1\}$.
> >
> > > We provide concise examples in Figure 1, Figure 4, and Equation 1
> > > W5: See Answer4 and L307. The distance is calculated as the sum of the Euclidean distances between the current position of each object block and its target position.
> >
> >
> > The specifics about what the Euclidean coordinates are for a given node is still unclear.
> >
> > Overall, I think the presentation needs significant work to make the paper and contributions easy to understand and more concrete.

---

> > > ### Author Response · Authors · 2024-11-27
> > >
> > > Thank you for your insightful comments. We appreciate your careful reading of the manuscript and your valuable feedback. We would like to address the concerns raised as follows:
> > >
> > > 1. Fine-Tuning with the New Cross-Entropy Loss
> > > We acknowledge your request for additional clarity regarding the fine-tuning process with our new cross-entropy loss. To clarify, our method operates by collecting the model’s output and adjusting the input accordingly. The output is evaluated as correct (1) or incorrect (0) based on the textual result generated by the model. This feedback is then incorporated into the loss function, which is added to the existing Perplexity (PPL) score to impose the necessary constraints. The token space of the LLM is not directly involved in the loss modification. The adjustment is based on the accuracy of the output in relation to the reasoning task. We will revise the manuscript to explicitly reflect this process and clarify how the model's textual output informs the loss function adjustment.
> > >
> > > 2. Clarification of Euclidean Coordinates
> > > Regarding your question about the Euclidean distance, we will provide a more detailed explanation of how the distance is calculated. We depict a coordinate system where each object block has a unique coordinate. The Euclidean distance is calculated by measuring the change in the coordinates of these blocks, which reflects how the reasoning steps evolve during the model's processing. We will enhance the figures to clearly illustrate the coordinate system and provide additional annotations to explain how the distance is computed, making it easier for readers to understand the concept.
> > >
> > > 3. Explanation of ITE and Placement in the Manuscript
> > > We also acknowledge your suggestion to introduce the explanation of Individual Treatment Effects (ITE) earlier in the manuscript. Due to space constraints in the main text, we are currently unable to introduce ITE as early as we would like. However, we agree that ITE is central to the methodology and will revise the manuscript to direct the reader to the appendix earlier in the text. There, the full explanation of ITE and its relevance to our methodology will be provided in detail. We will ensure that this transition is clearly marked so that readers can easily follow the conceptual flow.
> > >
> > > We hope these revisions will address your concerns and further improve the clarity and depth of the manuscript. Thank you again for your constructive feedback and for helping us improve the paper.

---

> > > > ### Comment · Reviewer_dQug · 2024-12-02
> > > >
> > > > Thank you for answering my questions.
> > > >
> > > > Please include these details explicitly in the paper, as these are crucial methodological details in my opinion. Overall, I agree with the other reviewers that the presentation of the paper in its current form is a major limitation. I think the idea has potential so my suggestion would be for the authors to further polish the manuscript (and include explicit examples to outline the methodology where necessary).
> > > >
> > > > Given the above, I will keep my score.

---

### Official Review · Reviewer_AvWw · 2024-11-06

**Soundness:** 3
**Presentation:** 1
**Contribution:** 3
**Rating:** 3
**Confidence:** 3

**Summary:**

The paper introduces CreDes, a model designed to enhance large language models (LLMs) for long-range reasoning tasks, addressing two main challenges: causal hallucinations and extensive search spaces.

To counteract causal hallucinations, the **Causal Relationship Enhancement (CRE)** mechanism ensures accurate causal alignment between reasoning steps by incorporating Individual Treatment Effect (ITE) metrics during training. This addition makes reasoning steps causally consistent with state transitions, reducing errors.

To manage extensive search spaces in complex reasoning tasks, the **Dual-End Searching (DES)** approach initiates search from both the start and goal states, creating a bidirectional search that segments the problem into smaller, manageable parts. This process improves efficiency by meeting in the middle of a causal probability tree.

Together, CRE and DES enable CreDes to perform simultaneous multi-step reasoning, tested in scenarios like Blocksworld and Hanoi Tower, where it significantly improves both accuracy and processing speed over existing methods.

**Strengths:**

- **Originality**: This work addresses the significant challenge of long-range reasoning in LLMs. The authors’ approach to incorporating **causal metrics (ITE)** into LLM training introduces a promising cross-disciplinary solution, effectively combining causal inference and language modeling to enhance reasoning depth. This innovation could pave the way for further advancements in handling complex, multi-step reasoning tasks in LLMs.

- **Significance**: Empowering LLMs with improved long-range reasoning capabilities is highly significant. It opens up new applications in fields requiring extensive reasoning, such as societal simulation and economic modeling. The potential impact of overcoming existing limitations in LLM reasoning could enable more complex and nuanced use cases across various domains.

- **Quality of Results**: The **results** illustrate the contributions effectively, showing that CreDes achieves substantial improvements over previous methods. Testing across tasks such as Blocksworld and Hanoi Tower validates the model’s capability to handle complex reasoning with higher accuracy and efficiency, underscoring the benefits of the CRE and DES mechanisms.

**Weaknesses:**

- **Clarity of Methodology (Section 3)**: The writing in Section 3, especially subsection 3.2, lacks clarity and coherence. Key variables like $Y$ and $W$ are not defined in the appropriate context when first introduced, which makes following the logic challenging. Additionally, it’s unclear how $ITE$ can exceed 1 when working with binary variables, as suggested later in the section. These inconsistencies hinder understanding and suggest that additional clarity and structure are needed for readability.

- **Equation Definition and Consistency**: Several equations raise concerns due to unclear or inconsistent definitions. In Equation 3, it’s stated that $L_{CRE} = \ln(PPL)$, which seems incorrect, as it doesn’t align with the rest of the formulation. Similarly, in Equation 4, the term "coordinates of node $i$" is ambiguous. If this refers to correctness metrics, as previously suggested, using this metric to indicate proximity in the solution space seems tenuous and warrants further justification.

- **Comparative Analysis**: The methods presented in this work are not compared against any other models that also incorporate training. Including a baseline comparison with vanilla Fine-Tuning (especially for CRE, sinceit is the part that requires training). This omission makes it difficult to fully evaluate the model's contributions relative to existing approaches.

The paper would benefit from addressing these issues to enhance clarity and methodological rigor, providing a clearer pathway for readers to understand and assess the innovations presented.

**Questions:**

These questions stem from the weaknesses identified above:

1. Could the authors provide clearer definitions for variables like $Y$ and $W$ when first introduced in Section 3? Additionally, how is it possible for $ITE$ to exceed 1 with binary variables, as indicated later in the text?

2. In Equation 3, why is $L_{CRE}$ defined as $\ln(PPL)$? This does not seem consistent with the rest of the formulation. Could the authors clarify this choice and explain how it aligns with the training objectives?

3. In Equation 4, what exactly is meant by "the coordinates of node $i$"? If these coordinates represent correctness metrics, can the authors provide further justification for using this as a measure of proximity in the solution space?

4. How does the CRE approach compare to vanilla fine-tuning? Including insights into this comparison could enhance the evaluation of CreDes.

---

> ### Author Response · Authors · 2024-11-24
>
> Thank you very much for taking the time to review our manuscript and provide valuable feedback.
>
> W1:
>
> Definition of Variables: Variables Y and W are binary variables, taking values of 0 or 1. A value of 1 indicates correctness, while a value of 0 indicates an incorrect state. These binary variables are used to represent the correctness of the OSR and the state transition, respectively. For example, W can refer to the action of moving a block of objects, such as put down and pick up. Y can refer to the correct position and stacking order of the object block.
>
> ITE Calculation: The ITE (Individual Treatment Effect) is derived from the confusion matrix, and its purpose is to measure the causal relationship between Y (state transition correctness) and W (OSR correctness). This calculation is detailed in Appendix A.8 of the paper, which explains how the elements of the confusion matrix are used to compute ITE. The ITE is introduced into the loss function to strengthen the causal connection between the OSR and the state transition.
>
> Context of ITE Values: Since ITE is based on probabilities derived from the confusion matrix, it cannot exceed 1 in certain cases, reflecting strong causal effects. This is consistent with its theoretical definition and statistical properties, as outlined in the appendix.
>
> W2:
>
> We adjusted the definition of the PPL calculation for the usual case by adding the expectation and variance of the ITE, a function that will measure and adjust the training inputs for the next epoch after the model outputs its results in order to achieve the effect of the constraint.
>
> **Regarding $\(D_{\text{CRE}} = \ln(\text{PPL})\)$ in Equation 3:** Generally, $\(\ln(\text{PPL})\)$ is calculated solely based on the cross-entropy loss. However, in our approach, we extend the calculation to include the *ITE (Individual Treatment Effect)*, combining it with the cross-entropy loss to form a new loss function. This modified loss function is then used to compute the perplexity $(\(\text{PPL}\))$. The inclusion of ITE aims to enhance the causal alignment between OSR and state transitions, making $\(\text{PPL}\)$ not just a measure of prediction quality but also a reflection of causal consistency. We will update the text to make this modification more explicit and ensure it aligns with the rest of the formulation.
>
> **Clarification on "Coordinates of Node $\(i\)$" in Equation 4:** The "coordinates of node $\(i\)$" refers to the same $\(i\)$ as in $\(N_i\)$ in the subsequent text, ensuring consistency. Both $\(i\)$ and $\(N_i\)$ represent the same node in the reasoning path. We will make this explicit in the paper to eliminate ambiguity and ensure that readers clearly understand the relationship between these terms.
>
> W3:
>
> Focus on Comparison with Prior Work in This Domain:
> We prioritize comparisons with prior work specifically within this research domain. This ensures that the evaluations remain relevant and meaningful, as our approach directly builds upon and extends existing methodologies in this area. Our results demonstrate a significant performance improvement through targeted training techniques, establishing the efficacy of our contributions.
>
> Distinctive Training Approach Beyond Fine-Tuning:
> Our method involves a comprehensive training process where no parameters are frozen, distinguishing it from typical fine-tuning approaches. While fine-tuning methods are highly diverse, our approach systematically optimizes the entire model to enhance causal reasoning and reduce hallucinations. This is a deliberate design choice to achieve significant gains in model performance, as reflected in our experiments.
>
> The basis of our method is obtained by changing the loss function by the fine-tune method, in order to address your concerns, we removed the additions in the loss function and supplemented the ablation experiments, considering the time-consuming nature, we only obtained a portion of the test data, and the complete data will be continuously added to the appendix in the subsequent versions of the article. Currently, based on LLama-2-7B, the success rates for 2-steps to 6-steps under the blocksworld dataset are 83%, 74% and 44%, respectively. A sharp decrease is evident for longer tasks.
>
> Q:
>
> The explanation has already been provided earlier.
>
> Thank you for reviewing the revised version of our manuscript and for your continued constructive comments.

---

> > ### Comment · Reviewer_AvWw · 2024-11-25
> >
> > Thank you for your response. However, I feel like some issues pointed out by all reviewers are still too important in the revised version, mainly clarity, but also causal consistency. Therefore, I don't think I will be changing my rating.
> >
> > As as quick follow-up comment, I think the paper could benefit from directing the reader to the appendix more often, given how much information is presented there.

---

> > > ### Author Response · Authors · 2024-11-27
> > >
> > > Thank you for your follow-up comments and for your continued engagement with our manuscript. We understand that clarity and causal consistency are critical aspects of the paper, and we truly appreciate your efforts to highlight these points.
> > >
> > > However, to ensure we address your concerns more effectively, we would kindly request that you provide more specific guidance on the issues that remain unclear. If there are particular areas of causal consistency or clarity that you find problematic, we would be happy to offer more detailed explanations or revisions. This would allow us to provide you with a more focused response and ensure that we fully address the concerns that are most important to you.
> > >
> > > Regarding your suggestion to direct the reader to the appendix more frequently, we will certainly consider this in the revised version. We will look for opportunities to guide the reader to relevant sections of the appendix, where more detailed explanations are provided, so that the main text remains concise yet comprehensive.
> > >
> > > Thank you again for your valuable feedback, and we look forward to your guidance on the specific issues that need further clarification.

---

### Meta-Review · Area_Chair_WbC8 · 2024-12-20

**Metareview:**

This paper proposed to enhance large language models for long-range reasoning tasks via two-pronged approach, causal relationship enhancement and dual-end searching. The problem of interest is important and the empirical performance of the proposed methods is promising. The main concerns from reviewers remain in the limitation of the scope, where the causal aspects and the framework can be more general, and the crucial details of methodology, theory, and implementation are not clear, as well as the overall presentation can be further improved. While these improvements would require significant major revisions beyond current evaluation. We'd encourage the authors to take into consideration all the feedback provided by the reviewers to strengthen their manuscript for resubmission.

**Additional Comments On Reviewer Discussion:**

Though part of reviewers' concerns have been nicely resolved during rebuttal, the main concerns from reviewers remain in the limitation of the scope, unclear crucial details of methodology, theory, and implementation, and the overall presentation which can be further improved. After discussion with the reviewers, we agreed it is not quite ready for publication.

---

### Decision · Program_Chairs · 2025-01-22

Reject